# Experimental Investigation and Optimization of the Semisolid Multicavity Squeeze Casting Process for Wrought Aluminum Alloy Scroll

**DOI:** 10.3390/ma13225278

**Published:** 2020-11-21

**Authors:** Yi Guo, Yongfei Wang, Shengdun Zhao

**Affiliations:** 1School of Energy and Power Engineering, Xi’an Jiaotong University, Xi’an 710049, China; yiguo666@mail.xjtu.edu.cn; 2School of Mechanical Engineering, Xi’an Jiaotong University, Xi’an 710049, China; sdzhao@mail.xjtu.edu.cn; 3State Key Laboratory of Fluid Power and Mechatronic Systems, Zhejiang University, Hangzhou 310027, China; 4State Key Laboratory of Materials Processing and Die and Mould Technology, Huazhong University of Science and Technology, Wuhan 430074, China

**Keywords:** wrought aluminum alloy, scroll compressor, semisolid forming process, microstructure, mechanical property

## Abstract

Scroll compressors are popularly applied in air-conditioning systems. The conventional fabrication process causes gas and shrinkage porosity in the scroll. In this paper, the electromagnetic stirring (EMS)-based semisolid multicavity squeeze casting (SMSC) process is proposed for effectively manufacturing wrought aluminum alloy scrolls. Insulation temperature, squeeze pressure, and the treatment of the micromorphology and mechanical properties of the scroll were investigated experimentally. It was found that reducing the insulation temperature can decrease the grain size, increase the shape factor, and improve mechanical properties. The minimum grain size was found as 111 ± 3 μm at the insulation temperature of 595 °C. The maximum tensile strength, yield strength, and hardness were observed as 386 ± 8 MPa, 228 ± 5 MPa, and 117 ± 5 HV, respectively, at the squeeze pressure of 100 MPa. The tensile strength and hardness of the scroll could be improved, and the elongation was reduced by the T6 heat treatment. The optimal process parameters are recommended at an insulation temperature in the range of 595–600 °C and a squeeze pressure of 100 MPa. Under the optimal process parameters, scroll casting was completely filled, and there was no obvious shrinkage defect observed inside. Its microstructure is composed of fine and spherical grains.

## 1. Introduction

Scroll compressors are popularly applied in air-conditioning systems because of numerous benefits, including high volumetric efficiency, low noise, low leakage, and low vibrations [1,2]. Due to excellent compression performance, scroll compressors are also recommended for use in fuel cell systems [3] or the energy storage field [4]. The main and key parts of scroll compressors are two interfitting scrolls, which form a positive displacement and compress the working fluid by rotating [5]. The scroll-forming method is crucial for the manufacturing and production of scroll compressors because of the complex shape with involute curves [6]. Hot forging and machining processes are generally used to fabricate scrolls for a typical refrigeration scroll compressor. However, some drawbacks are observed during these manufacturing processes, which include low efficiency, high production cost, and low material usage coefficient [7,8,9]. High-pressure die casting has also been applied for the fabrication of scrolls, but the turbulent flow in this process causes gas and shrinkage porosity [10] because of the high filling rate [11].

Gas and shrinkage porosity can be effectively eliminated by squeeze casting technology, which has been recommended to manufacture the complex component from aluminum alloys and aluminum metal–matrix composites [12,13,14]. The semisolid squeeze casting (SSSC) process, which combines the excellent filling performance of a semisolid metal with the characteristics of squeeze casting technology [15,16], is one of the most widely used semisolid metal-forming (SSMF) techniques. It integrates the easy fluidity of the liquid casting process and the high mechanical properties of solid-state plastic forming and is considered one of the most promising metal processing technologies of the 21st century [17,18,19]. In the SSSC process, a semisolid slurry is first injected into a mold cavity. The rheological filling, crystallization solidification, rheological feeding, and a small amount of plastic deformation are then performed under pressure. The high-strength product with a dense interior, accurate size, and few macro- and microdefects can finally be obtained after the SSSC process [20].

In recent decades, research on the SSSC process has drawn the attention of scholars. Lü et al. [21] investigated the effects of the SSSC process on the microstructure and mechanical properties of the long period stacking ordered (LPSO)-structure-reinforced Mg_97_Zn_1_Y_2_ alloy. It was found that the structure of LPSO and α-Mg grains in the studied alloy can be dramatically improved by the SSSC process. Wang et al. [22] proposed SSSC to fabricate ZL104-alloy-connecting rods. The influences of remelting temperature, mold temperature, and squeeze pressure on the mechanical properties of the connecting rod were investigated. It was concluded that tensile strength and elongation were increased by 22% and 17%, respectively, through the SSSC process. Zhai et al. [23] studied the structure evolution and mechanical properties of the ZL109 alloy prepared by the SSSC process. It was revealed that the sphericity of the α phase and the refining of the silicon phase could be obtained due to the friction between solid and liquid. The elongation of the ZL109 alloy rose to 1.42% when heat treatment was performed after the SSSC process. Masoumi et al. [24] studied the influence of applied pressure on the microstructure and tensile properties of the Mg-Al-Ca alloy. They pointed out the tensile properties of casting increased with increased applied pressures, which can be attributed to the casting densification and presence of a high amount of solute in the matrix.

The preparation of a semisolid billet is an indispensable step for the SSSC process. Electromagnetic stirring (EMS) is a contactless technique commonly applied to produce forced convection and adjust fluid flow during the solidification of the metal melt, which has become one of the main methods for the production of semisolid slurry or billets [25,26]. EMS works similar to the induction motor, which contains a stator and a rotor [27]. When the three-phase alternating current is supplied to the windings of the stator, the induced magnetic field is produced in this field. The induction current is obtained when the rotor cuts the induced magnetic field. When the rotor is replaced by the melt metal bulk, EMS initiation is achieved. Compared to traditional stirring, several benefits to flow control and grain refinements, such as nonoxidation [28], nonpollution [29], continuous production [30], and easy operation [31], can be obtained by the EMS technique.

Due to the drawbacks of the conventional process method and the advantages of the SSSC and EMS techniques, the EMS-based semisolid multicavity squeeze casting (SMSC) process is proposed in this paper for manufacturing aluminum alloy scrolls. This work aims to effectively improve the production efficiency, microstructure, and mechanical properties of the scroll. Firstly, the near-final forming of the scroll can be realized via the processed process, significantly improving the material utilization rate. Secondly, the concept of one mold for four castings can be verified, visibly improving the production efficiency. Thirdly, the semisolid billet with fine and spherical grains can be prepared by the EMS process, improving the mechanical properties of the scroll. The effects of insulation temperature, squeeze pressure, and the heat treatment of the micromorphology and the mechanical properties of the scroll were investigated experimentally, based on which optimal process parameters were obtained for the manufacturing of scrolls by the EMS-based SMSC process.

## 2. Materials and Methods

The commercial wrought aluminum alloy 2A50 (Aluminum Corporation of China Limited, Beijing, China) was used as the experimental material in this work. Its chemical components are summarized in Table 1, which shows that the main alloy element is copper. The solidus and liquidus temperatures of the studied material are 521 °C and 615 °C, respectively. This means that the semisolid temperature range is as wide as up to 94 °C, which is suitable for semisolid casting. Information on the structure and dimensions of the studied scroll for a typical scroll compressor applied in the automotive air conditioning is shown in Figure 1.

The EMS-based SMSC process of one die with four aluminum alloy scrolls proposed in this paper mainly includes four stages, as shown in Figure 2. (A) Stage I: The 2A50 aluminum alloy semisolid billet is prepared by electromagnetic stirring. The pouring temperature of the molten metal is 645 °C, and the preheating temperature of the stirring chamber is 350 °C. The winding current and the electrical frequency of the EMS are set as 30 A and 30 Hz, respectively. (B) Stage II: The second remelting of the semisolid billet is performed at a set insulation temperature for 15 min. (C) Stage III: The semisolid multicavity squeeze forming of the scroll is performed under the designed pressure by using a multicavity squeeze casting hydraulic press (Xi’an Jiaotong University, Xi’an, China), as shown in Figure 3a, and a multicavity squeeze forming mold, as shown in Figure 3b, where the preheating temperature of the mold is 350 °C. (D) Stage IV: T6 heat treatment is performed on the formed scrolls, which mainly includes a solution treatment and aging treatment. This stage is performed at a solution temperature of 495 °C for 5 h and an aging temperature of 180 °C for 8 h.

In this paper, 8 multicavity squeeze experiments were performed to investigate the effects of the operating parameters on the microstructure evolution of the scrolls. As shown in Table 2, the operating parameters studied in this paper contain the insulation temperature in Stage II and the squeeze pressure in Stage III. In order to compare and analyze the advantages of the semisolid forming process compared with the near-liquidus and liquid-forming process, a liquid temperature of 645 °C, a near-liquidus temperature of 615 °C, and semisolid temperatures of 600 and 595 °C were designed in Experiments 1–4, respectively, to study the effect of insulation temperature on the microstructures. The squeeze pressure varied from 25 to 125 MPa to reveal the effects of squeeze pressure on the microstructures. In Experiments 1–4, the squeeze pressure was set at 100 Mpa with variation in the insulation temperature, whereas in Experiments 4–8, the insulation temperature was set at 595 °C with a change in the squeeze pressure. It is worthy to note that the insulation temperature in Experiments 5–8 was designed based on the results from Experiments 1–4 to only investigate the effects of different squeeze pressures without the variation in the insulation temperature. All 8 experiments were performed with the T6 heat treatment.

A schematic diagram of the whole semisolid multicavity squeeze process for fabricating four scrolls under one mold is illustrated in Figure 4. Under the action of the squeeze head, the semisolid slurry flows from the vertical and horizontal runner to the ingate runner and fills the mold cavity. The gas and redundant semisolid slurry flow out from the overflow trough. The scrolls can then be fabricated after the solidification of semisolid slurry. Through this designed mold, symmetry in the filling flow of the semisolid materials for the four scrolls can be achieved in the squeeze process. 

Considering the symmetry of the proposed semisolid multicavity squeeze method, one of the four scrolls was sampled for the investigation in this work. Three samples from the scrolls processed at different operating parameters were collected to study the microstructures. These samples were located at the inner gate (Position A1), the central region of the scroll (Position A2), and the bottom region of the scroll (Position A3), as shown in Figure 4. The microstructure of the samples was observed by an optical microscope (NIKON ECLIPSE LV 150N, Nikon, Tokyo, Japan) from 10 different fields of view for each sample, based on which the average grain diameter and shape factor of the metallographic structure were firstly measured using the Image-Pro Plus 6.0 software (Media Cybernetics, Rockville, MD, USA) and quantitatively analyzed by Equations (1) and (2) [32], respectively. Samples from the central region of the scroll (Position A2) were used for testing the mechanical properties by using a universal materials tester (INSTRON 5982, Instron, Norfolk, MA, USA) and a hardness tester (HV-1000, Shanghai Shangcai Testermachine Co.,Ltd, Shanghai, China). For obtaining the tensile strength, yield strength, and elongation, samples were stretched to break at the speed of 2 mm/min. In order to test the hardness, samples were processed under a test load of 200 g with a pressure holding time of 15 s. The error of all the experimental data in this work was represented by the standard deviation.
(1)$$D=\frac{\sum_{N=1}^{N}\sqrt{4A/\pi }}{N},$$
(2)$$F=\frac{\sum_{N=1}^{N}4\pi A/P^{2}}{N},$$
where *D* is the average equivalent diameter of the crystal grains, *A* is the area of the crystal grain, *P* is the circumference of the crystal grain, *N* is the number of the crystal grains, and *F* is the average shape factor of the crystal grains. 

## 3. Results and Discussion

### 3.1. Effects of Insulation Temperature on the Microstructures and Mechanical Properties

The microstructures of the sample collected from Position A2 formed at liquid temperature (645 °C), near-liquidus temperature (615 °C), and semisolid temperatures (600 and 595 °C) are shown in Figure 5. It was found that the grain size significantly reduced, and the morphology improved when the insulation temperature dropped from the liquid state to a semisolid value. It is shown in Figure 5a that the morphology of crystal grains was extremely poor when Stage II was performed at a liquid temperature, where most of the coarse dendrites were more than 200 μm in length. Compared to the liquid-forming process, it was found that the grain morphology evolved to the rose shape, and the grain size was obviously reduced when Stage II was performed at the near-liquidus temperature, which is presented in Figure 5b. When Stage II was performed at the semisolid temperatures, the microstructure was observed with a round morphology and small sizes.

The effects of the insulation temperature in Stage II on the average grain size and shape factor are summarized in Figure 6. A general increase in the average size and a general reduction in the shape factor were found when the insulation temperature varied from 595 to 645 °C. A rapid increase in the average grain size from 119 ± 4 to 238 ± 24 μm was observed, whereas the shape factor significantly decreased from 0.62 ± 0.06 to 0.54 ± 0.09 when the insulation temperature increased from 600 to 615 °C. A slight increase in the average grain size (from 238 ± 24 to 288 ± 32 μm) and a slight drop in the shape factor (from 0.54 ± 0.09 to 0.52 ± 0.09) were observed with the increasing of insulation temperature from 615 to 645 °C. This was because when the degree of the superheat was reduced near the liquidus temperature, the nucleation rate and grain proliferation increased while the thick dendritic arms were fragmented during the filling and flow process. The morphology of crystal grains therefore gradually changed from liquid-formed coarse dendrites to rose-like crystal grains. In terms of the semisolid forming, in other words, the insulation temperature varied in the semisolid temperature range, reducing the insulation temperature of the blank can reduce the coarsening time of the crystal grains during the filling process and effectively limit the excessive growth of the crystal grains to ensure the crystal grains have a smaller size and a relatively high roundness. 

The mechanical properties investigated in this study are tensile strength, yield strength, elongation, and hardness. The mechanical properties of the scroll after the T6 heat treatment at different insulation temperatures in Stage II are presented in Figure 7. It was found that the tensile strength, yield strength, elongation, and hardness increased and almost stabilized when the insulation temperature dropped from the liquid state to the semisolid temperature range. The tensile strength increased from 272 ± 25 to 385 ± 9 MPa when the insulation temperature decreased from 645 to 600 °C, which further rose to 386 ± 8 MPa at the insulation temperature of 595 °C. The elongation of the scroll was measured as 5.4 ± 1.2% at the insulation temperature of 645 °C, which significantly increased to 7.5 ± 0.5% and 7.6 ± 0.5% at the insulation temperatures of 600 and 595 °C, respectively. An increase in the yield strength was observed from 163 ± 13 MPa at the insulation temperatures of 645 °C to 228 ± 5 MPa and 595 °C. The hardness of the scroll was observed as 85 ± 9 and 95 ± 8 HV at the insulation temperatures of 645 and 615 °C, respectively, which dramatically increased up to 117 ± 5 HV at the insulation temperature of 595 °C. This improvement in the mechanical property of the scroll was because the primary crystal grains with uniform size and round shape can be obtained at the semisolid temperature. These improved crystal grains can effectively reduce the microscopic stress concentration generated by the coarse dendrites formed at the liquid-state temperature and slow down the formation and expansion of voids. Moreover, when the insulation temperature was relatively high at the liquidus or near-liquidus temperatures, the turbulent jetting of the slurry would take place during the filling process due to the low viscosity, which would cause the internal defects inside the scroll and the consequently low mechanical properties. The improvement in the yield strength with the decrease in the insulation temperature was because of the reduced grain size, which can be expressed by the classical Hall–Petch relationship, as shown in Equation (3) [33,34]
(3)$$\sigma_{y}=\sigma_{0}+k_{y}d^{-1/2},$$
where $\sigma_{0}$ is the intrinsic lattice strength, *d* is the average grain size and $k_{y}$ is the Hall–Petch coefficient.

### 3.2. Effects of Squeeze Pressure on the Microstructures and Mechanical Properties

According to the analysis in Section 3.2, it was found that the microstructure and mechanical properties of the scroll were between 595 and 600 °C. Therefore, the insulation temperature was designed as 595 °C to investigate the effects of the squeeze pressure without the variation in the insulation temperature for the purpose of energy saving. The microstructures of the scroll at the central region prepared by different squeeze pressures from 25 to 125 MPa in Stage III are shown in Figure 8. As shown in Figure 8a, under the squeeze pressure of 25 MPa, the crystal grains are round and uniform. When the squeeze pressure increased to 50 MPa, the grain size with the relatively round grain morphology slightly reduced, as presented in Figure 8b. However, the grain distribution became dense, with some grains coming into contact with each other, resulting in blurred grain boundaries and the slightly deteriorated grain morphology when the squeeze pressure increased up to 75 Mpa, which is presented in Figure 8c. When the squeeze pressure further increased to 125 MPa, a lot of crystal grains were adhered to and merged with each other. The newly merged crystal grains have different morphologies. During the high-pressure filling process, grains were coarsened to form a large number of grain clusters with fuzzy grain boundaries, which caused the large size grains and morphology deterioration.

The effects of the squeeze pressure on the average grain size and shape factor of the scroll are presented in Figure 9. It was found that the average grain size slightly decreased from 116 ± 4 to 112 ± 3 μm when the squeeze pressure increased from 25 to 50 MPa. After that, with the increase in the squeeze pressure in Stage III, the average grain size slightly grew up to 113 ± 3 μm at the squeeze pressure of 75 MPa and rapidly increased to 130 ± 4 μm at the squeeze pressure of 125 MPa. The shape factor showed a significant decrease from 0.74 ± 0.05 to 0.60 ± 0.07 when the squeeze pressure varied from 25 to 75 MPa. It had a slight increase when the squeeze pressure was 100 MPa but further reduced to 0.54 ± 0.08 with the squeeze pressure of 125 MPa.

In the process of semisolid squeeze casting, the liquidus temperature of the metal slightly increases, and the cooling for the slurry is improved with the increase in the squeeze pressure, which consequently influences the microstructure of the scroll. The influence of squeeze pressure on the liquidus line of the alloy can be illustrated by the Clausius–Clapeyron equation, as shown in Equation (4) [35].
(4)$$\frac{\Delta T_{f}}{\Delta P}=\frac{T_{f}\left(V_{l}-V_{s}\right)}{\Delta H_{f}},$$
where $T_{f}$ is the equilibrium solidification temperature, K; $V_{l}$ is the volumetric specific heat capacity of liquid metal, J/(kg·K); $V_{s}$ is the volumetric specific heat capacity of solid metal, J/(kg·K); $\Delta H_{f}$ is the latent heat of metal fusion, J/g; *P* is the squeeze pressure, MPa.

The results of Figure 9 can be explained by the Clausius–Clapeyron equation. According to Equation (4), the liquidus temperature of the slurry can increase by the enhancement in the squeeze pressure, which consequently increases the degree of subcooling of the slurry and prolongs the grain coarsening time. The slurry was forced to closely contact the surface of the mold by the increase in the squeeze pressure, which improved the heat transfer between the slurry and the mold, and therefore, the α-Al grains did not have enough time to grow. The grains were forced to undergo a certain compression or elongation deformation to adapt to the filling movement of the slurry by the squeeze pressure, which caused a large number of grains compacted under the high pressure and therefore, the poor morphology and large size clusters were observed under the high squeeze pressure.

The mechanical properties of the scroll samples under different squeeze pressures after the T6 heat treatment are shown in Figure 10. The tensile strength increased from 341 ± 7 to 386 ± 8 MPa when the squeeze pressure increased from 25 to 100 MPa but slightly decreased to 383 ± 12 MPa at the squeeze pressure of 125 MPa. Similarly, a significant growing trend was observed for the elongation (from 4.5 ± 0.4% to 7.6 ± 0.5%) when the squeeze pressure was enlarged from 25 to 100 MPa. However, the elongation reduced to 7.5 ± 0.9% at the squeeze pressure of 125 MPa. The yield strength increased from 214 ± 5 MPa at the squeeze pressure of 25 MPa to 228 ± 5 MPa at the squeeze pressure of 100 MPa, which dramatically decreased to 212 ± 5 MPa when the semisolid billet was processed at the squeeze pressure of 125 MPa. This can be attributed to the change in the average grain size as the squeeze pressure rose. The hardness of the scroll sample increased from 98 ± 6 to 117 ± 5 HV when the squeeze pressure rose from 25 to 100 MPa, which remained almost stable under the squeeze pressure of 125 MPa. This was because the degree of compaction of the slurry continued to increase and gradually reached a stable value when the squeeze pressure increased. When it reached a certain value, a further increase in the squeeze pressure had no significant improvement in the degree of compaction, so its hardness did not continue to increase. Moreover, the adverse effect on the equipment would be obtained if the squeeze pressure is too high. Therefore, the squeeze pressure is recommended no more than 100 MPa during the semisolid multicavity squeeze process.

### 3.3. Effects of Heat Treatment on the Microstructures and Mechanical Properties

Based on the experimental results presented in Section 3.1 and Section 3.2, the optimal operating range of the parameters was identified as 595–600 °C for the insulation temperature in Stage II and 75–100 MPa for the squeeze pressure in Stage III. Therefore, the microstructure of the scroll was processed at the insulation temperature of 600 °C, and the squeeze pressure of 100 MPa after the T6 heat treatment was obtained for analysis, as shown in Figure 11. It was found that the solute atoms in the 2A50 supersaturated solid solution were continuously enriched along the grain boundaries or other crystal planes to form segregation zones and gradually transformed into stable second-phase particles distributed around the grain boundaries through the T6 heat treatment. Therefore, the grain boundaries of the microstructure are more obvious compared to the microstructure of the scroll without the heat treatment, as shown in Figure 5c.

A comparison of the mechanical properties between the scrolls before and after the T6 heat treatment is shown in Table 3. It was found that the hardness, tensile strength, and yield strength of the scroll after the T6 heat treatment were higher than those without the T6 treatment. The hardness of the scroll increased from 73 ± 4 HV (before the T6 heat treatment) to 117 ± 5 HV (after the T6 treatment). The tensile strength was enhanced to 386 ± 8 MPa, and the yield strength was increased to 228 ± 5 MPa through the T6 heat treatment. This was because a large number of strengthening phases were dispersed and precipitated evenly on the α-Al grain boundary after the aging treatment of the supersaturated solid solution. These brittle and hard dispersion-strengthening phases were ”pinned” on the grains or grain boundaries, which hindered the expansion of dislocation movement and increased the strength of the matrix [36]. Consequently, the tensile strength and the hardness were improved by the T6 heat treatment.

However, the elongation of the scroll decreased after the T6 heat treatment, which reduced from 12.3 ± 0.4% to 7.6 ± 0.5%. The partial dispersion-strengthening phase generated during the T6 heat treatment split the continuity of the microstructure. During the stretching process, although the α-Al matrix has high plasticity and undergoes the macroscopic deformation with the sample, the hard and brittle particles, such as CuAl_2_, have extremely poor plasticity, which is prone to stress concentration. This stress concentration worked as the crack source, which caused the fracture of the strengthening phase on the matrix and the fusion of the microcrack. Therefore, the toughness and elongation decreased after the T6 treatment.

### 3.4. Macromorphology and Microstructures of the Scroll under the Optimal Process Parameters

The macro morphology of the scroll obtained under the optimal process parameters (i.e., insulation temperature of 595 °C, squeeze pressure of 100 MPa) is shown in Figure 12. It was found that the scroll was fully filled, with clear outlines and smooth surfaces, without any obvious shrinkage, entrainment, cracks, mucosa, or any other common macro defects. The microstructure of different positions of the scroll labeled by A–F in Figure 12 is presented in Figure 13.

It can be found in Figure 13 that the microstructure of the scroll was mainly composed of fine and spherical crystal grains. Although the grain distribution throughout the scroll was relatively uniform, a slight difference in the crystal grain morphology between different positions of the scroll was observed. As shown in Figure 13a,b, a large number of crystal grains were elongated or squashed at the location close to the inner gate (Positions A and B) during the flow filling process as they were directly affected by the squeeze pressure, which formed grain clusters with obvious directionality as the slurry flowed. Figure 13c,d illustrates that part of the crystal grains in the central region of the scroll (Positions C and D) were elongated by contacting each other under the high-pressure condition. Some crystal grains even contacted and merged to form elongated crystal grains with irregular morphology. The obvious flow direction in the whole grain distribution can be observed in the microstructure of Positions C and D. The grains were spherical at the bottom region of the scroll as it received less pressure due to the attenuation of the squeeze pressure along with the casting, which is shown in Figure 13e,f. Moreover, there was no obvious deformation observed at Positions E and F.

## 4. Conclusions

This paper proposes the EMS-based SMSC process for manufacturing aluminum alloy scrolls. The insulation temperature, squeeze pressure, and heat treatment of the micromorphology and the mechanical properties of the scroll are investigated experimentally. The main conclusions can be drawn as follows:
(1)The microstructures of the scroll at the liquid, near-liquid, and semisolid insulation temperature are composed of coarse dendrites, rose-like crystal grains, and near-spherical crystal grains, respectively. Reducing the insulation temperature can significantly reduce the grain size, improve the grain morphology, and enhance the mechanical properties.(2)Reducing the insulation temperature to the semisolid range can decrease the grain size, increase the shape factor, and improve the mechanical properties. The minimum grain size was found as 111 ± 3 μm at the insulation temperature of 595 °C.(3)Increasing the squeeze pressure is beneficial to reduce the microscopic pores of the microstructure, increase the compactness of the crystal grains, and improve the mechanical properties. However, the improvement in the mechanical properties is not obvious when the squeeze pressure exceeds 100 MPa. The maximum tensile strength, yield strength, and hardness were observed as 386 ± 8 MPa, 228 ± 5 MPa, and 117 ± 5 HV, respectively, at the squeeze pressure of 100 MPa.(4)Although the tensile strength and hardness of the scroll can be improved by the T6 heat treatment, the elongation was reduced after the T6 heat treatment. The morphology of the grains is relatively round, and no obvious deformation was observed at the bottom part of the scroll casting due to the attenuation of the squeeze pressure.(5)Based on the experimental results, the optimal process parameters are recommended as an insulation temperature of 595–600 °C and a squeeze pressure of 100 MPa.(6)Under the optimal process parameters, the scroll casting is completely filled, and there is no obvious shrinkage defect observed inside. Its microstructure is composed of fine and spherical grains. Some of the grains in the central region of the scroll are deformed by contact showing obvious flow direction. A large number of grains at the top region of the scroll casting are elongated or flattened and formed a cluster of crystal grains with strong orientation under the direct action of the squeeze pressure.

## Figures and Tables

**Figure 1 materials-13-05278-f001:**
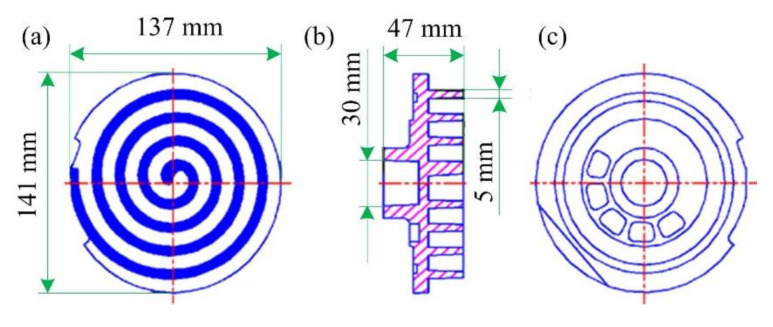
Structure and dimensions of the scroll for scroll compressors applied in automotive air conditioning: (**a**) front view, (**b**) sectional view, and (**c**) back view.

**Figure 2 materials-13-05278-f002:**
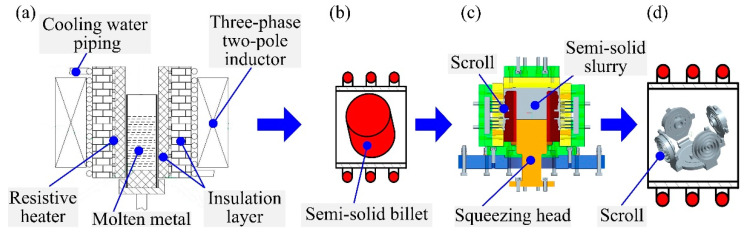
Electromagnetic stirring (EMS)-based semisolid multicavity (SMSC) process of one mold for four aluminum alloy scrolls: (**a**) Stage—EMS preparation, (**b**) Stage II—remelting, (**c**) Stage III—forming process, and (**d**) Stage IV—T6 heat treatment.

**Figure 3 materials-13-05278-f003:**
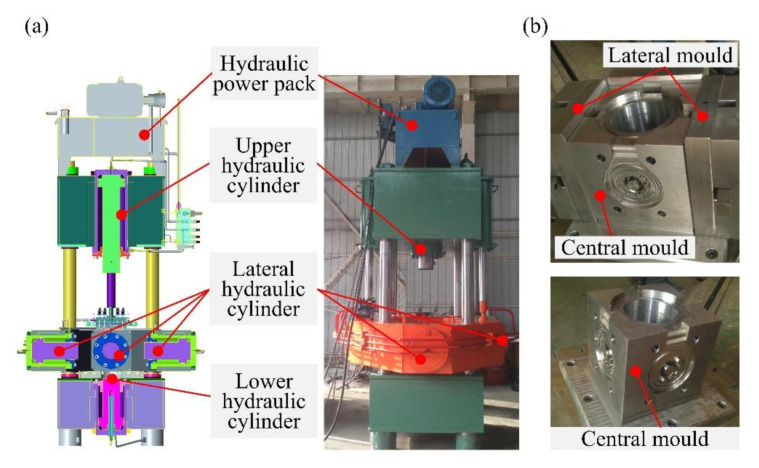
(**a**) The multicavity squeeze casting hydraulic press and (**b**) the multidirection squeeze forming mold for manufacturing scrolls.

**Figure 4 materials-13-05278-f004:**
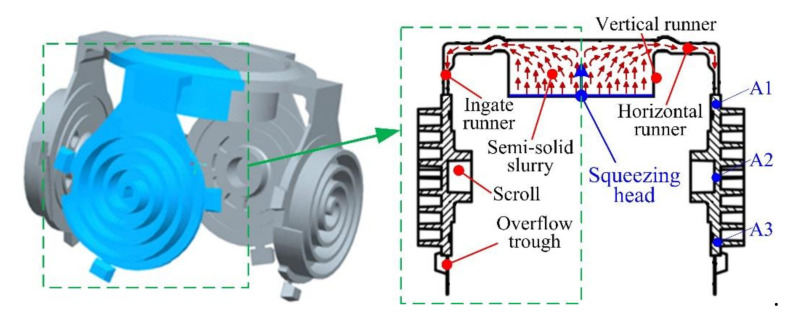
Schematic diagram of the whole SMSC process for fabricating four scrolls with one mold.

**Figure 5 materials-13-05278-f005:**
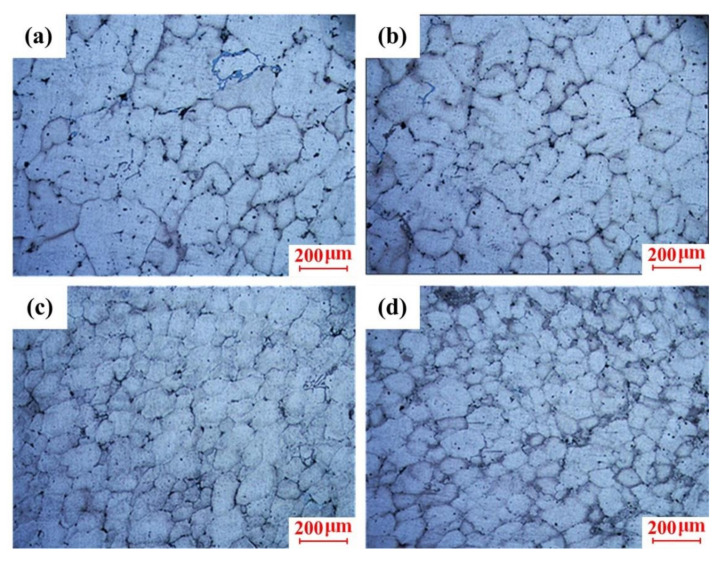
Microstructure of scroll under different holding temperature: (**a**) 645 °C, (**b**) 615 °C, (**c**) 600 °C, and (**d**) 595 °C.

**Figure 6 materials-13-05278-f006:**
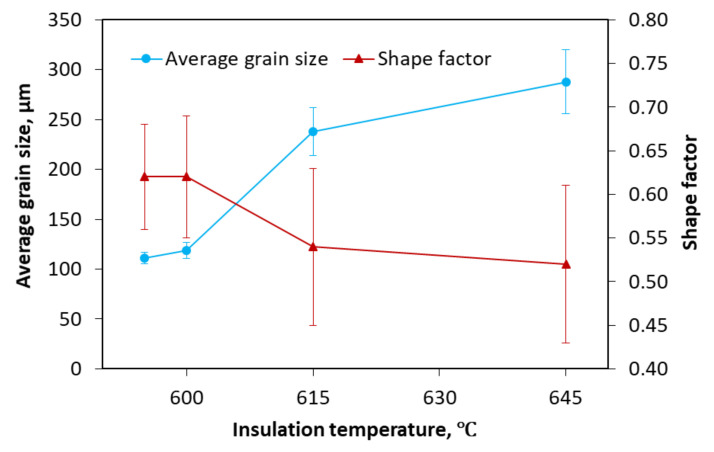
Effects of insulation temperature on the average grain size and shape factor.

**Figure 7 materials-13-05278-f007:**
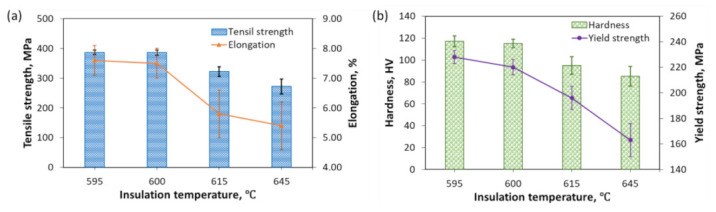
Effects of insulation temperature on the mechanical property of scrolls: (**a**) tensile strength and elongation; (**b**) yield strength and hardness.

**Figure 8 materials-13-05278-f008:**
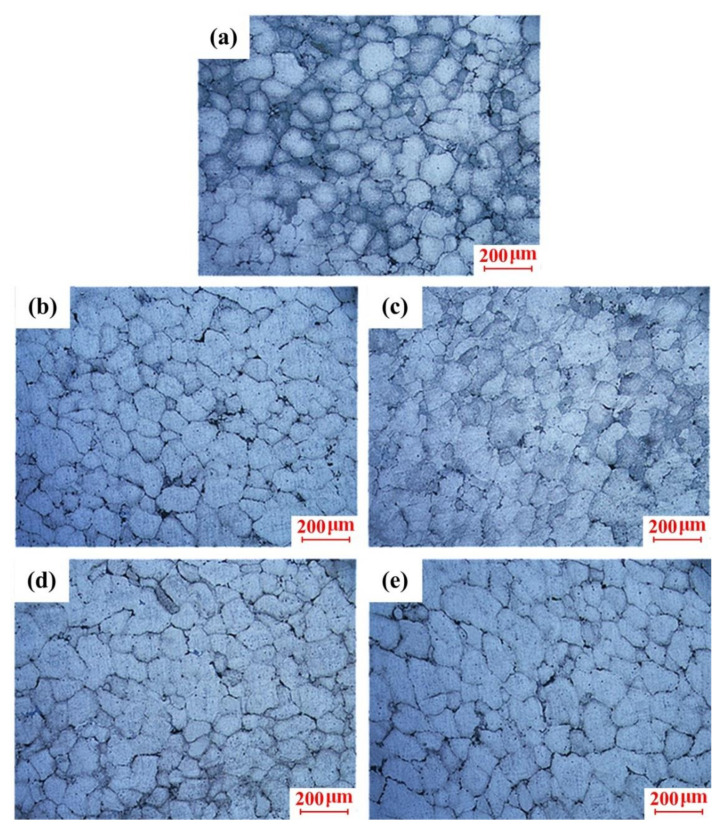
Microstructure of the scroll under different squeeze pressures: (**a**) 25 MPa, (**b**) 50 MPa, (**c**) 75 MPa, (**d**) 100 MPa, and (**e**) 125 MPa.

**Figure 9 materials-13-05278-f009:**
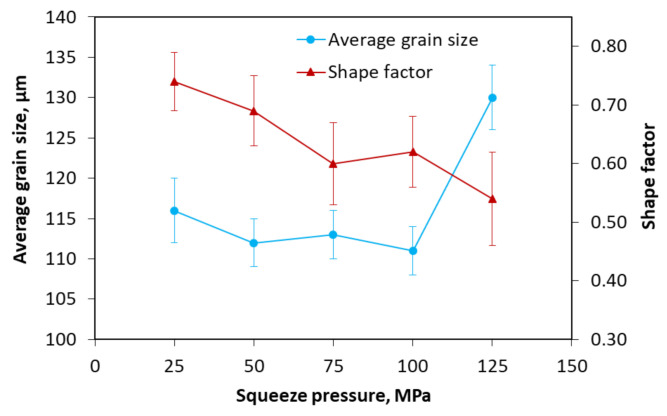
Effects of squeeze pressure on the average grain size and shape factor.

**Figure 10 materials-13-05278-f010:**
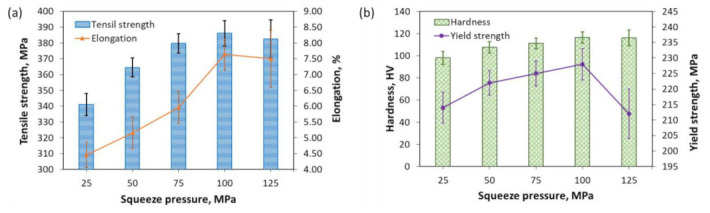
Effects of squeeze pressure on the mechanical property of scrolls: (**a**) tensile strength and elongation; (**b**) yield strength and hardness.

**Figure 11 materials-13-05278-f011:**
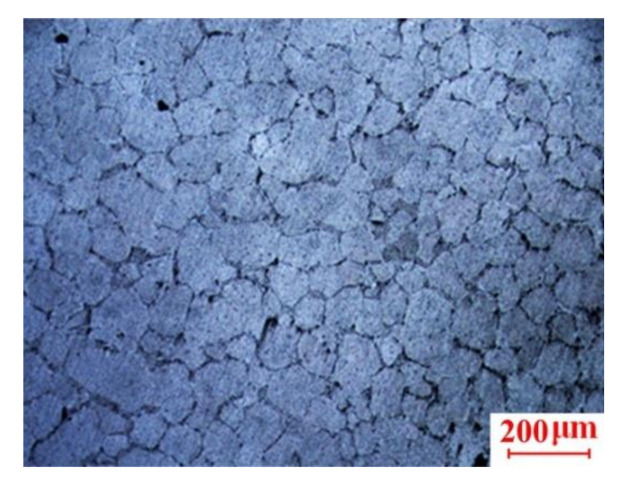
Microstructure of scrolls after the T6 heat treatment.

**Figure 12 materials-13-05278-f012:**
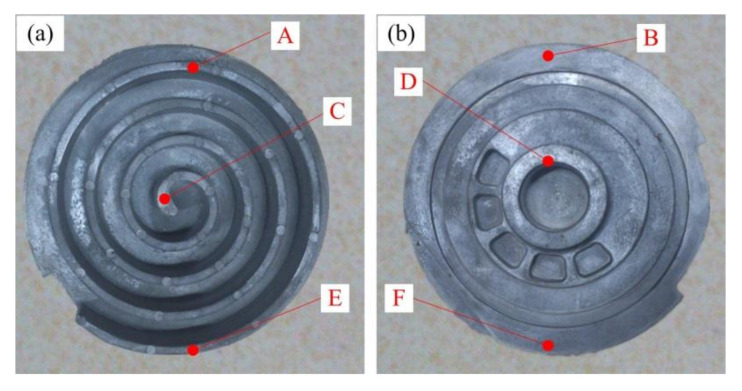
Macro morphology of the scroll: (**a**) front view and (**b**) back view.

**Figure 13 materials-13-05278-f013:**
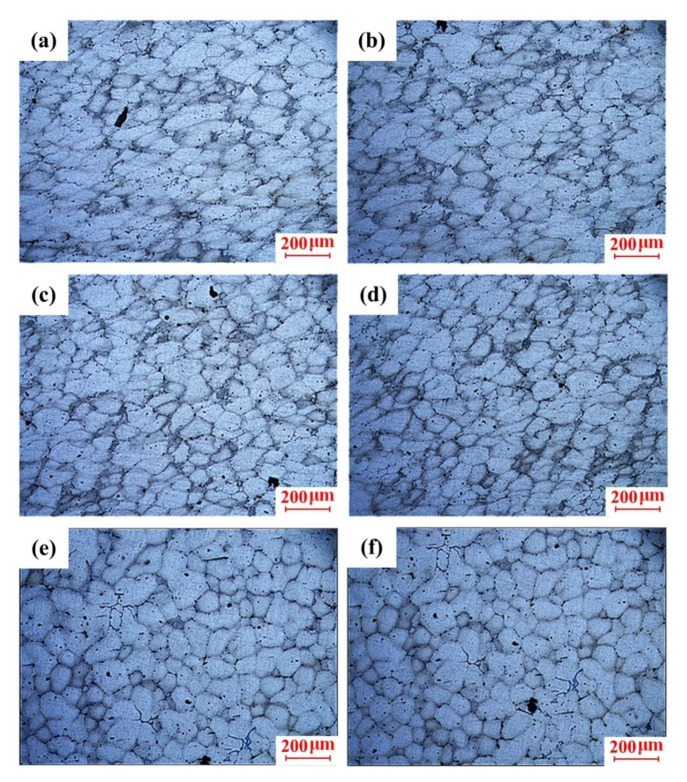
Microstructures at different positions of the scroll: (**a**) Position A in the top region, (**b**) Position B in the top region, (**c**) Position C in the central region, (**d**) Position D in the central region, (**e**) Position E in the bottom region, and (**f**) Position F in the bottom region.

**Table 1 materials-13-05278-t001:** Chemical composition of 2A50 aluminum alloy (wt.%).

Cu	Si	Mg	Mn	Zn	Ti	Ni	Fe	Al
2.43	0.82	0.68	0.53	0.12	0.06	0.05	0.01	Surplus

**Table 2 materials-13-05278-t002:** Experimental process parameters.

Experimental Number	Insulation Temperature (°C)	Squeeze Pressure (Mpa)	Heat Treatment
1	645	100	T6
2	615	100	T6
3	600	100	T6
4	595	100	T6
5	595	25	T6
6	595	50	T6
7	595	75	T6
8	595	125	T6

**Table 3 materials-13-05278-t003:** Mechanical properties of scrolls before and after the T6 heat treatment.

Heat Treatment	Hardness (HV)	Tensile Strength (MPa)	Yield Strength (MPa)	Elongation (%)
Before the T6 heat treatment	73 ± 4	286 ± 7	124 ± 6	12.3 ± 0.4
After the T6 heat treatment	117 ± 5	386 ± 8	228 ± 5	7.6 ± 0.5

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
