# Peer review of "Experimental Investigation and Optimization of the Semisolid Multicavity Squeeze Casting Process for Wrought Aluminum Alloy Scroll"

_materials, 2020, doi:10.3390/ma13225278_

Round 1

Reviewer 1 Report

In General, the article makes a positive impression. The conclusions are clearly formulated and the literature review does not raise questions. The research topic is relevant and should be of interest to a wide audience. But there are several points that need to be considered in more detail.
1. The aim of this work is not marked. Specify the purpose of the work more specifically.
2. Research methods are not fully disclosed in Chapter 2. How were the data obtained from Figures 6, 7, 9 and 10? What equipment was used?
3. The authors study the grain size of an aluminum alloy. It may be worth providing the microstructure at a higher magnification.
4. How was the tensile test performed? This must be specified in the article.
I believe that the article can be accepted after minor revision.

Author Response

The authors would like to take this opportunity to sincerely thank the reviewer for his/her valuable comments. We have studied all the comments carefully and have made corresponding corrections that we hope will meet with your approval. The detailed responses to the reviewer’s comments are provided below.

In General, the article makes a positive impression. The conclusions are clearly formulated and the literature review does not raise questions. The research topic is relevant and should be of interest to a wide audience. But there are several points that need to be considered in more detail.

  1. The aim of this work is not marked. Specify the purpose of the work more specifically.

Response: Thanks for the reviewer’s comment. The introduction has been revised to better specify the purpose of our work. The revised and added content is shown as follows.

This work aims to effectively improve production efficiency, microstructure, and mechanical properties of the scroll. Firstly, the near-final forming of the scroll can be realized via the processed process, significantly improving the material utilization rate. Secondly, the concept of one mould for four castings can be verified, obviously improving the production efficiency. Thirdly, the semi-solid billet with fine and spherical grains can be prepared by the EMS process, improving the mechanical properties of the scroll.

Please see lines 87-93 on page 2 in the revised manuscript.

  1. Research methods are not fully disclosed in Chapter 2. How were the data obtained from Figures 6, 7, 9 and 10? What equipment was used?

Response: Thanks for pointing this out. The method and relevant equipment for obtaining the data have been added in the revised manuscript. The added content is shown as follows.

The microstructure of the samples was observed by the optical microscope (NIKON ECLIPSE LV 150N) from 10 different fields of view for each sample, based on which the average grain diameter and shape factor of the metallographic structure are firstly measured using the Image-Pro Plus 6.0 software, and then quantitatively analysed by Equations (1) and (2) [32], respectively. Samples from the central region of the scroll (Position A2) were used for testing the mechanical properties by using the universal materials tester (INSTRON 5982) and the hardness tester (HV-1000).

Please see lines 159-165 on page 5.

  1. The authors study the grain size of an aluminum alloy. It may be worth providing the microstructure at a higher magnification.

Response: Thanks for the reviewer’s comment. We agree with the reviewer that the microstructure at a higher magnification is better for the study of the microstructure evolution. However, in this paper, the average grain size and shape factor are the main concerns in order to better reflect the uniform distribution of the semi-solid microstructure. Therefore, the microstructure photo with the magnification of 100 times but not the higher magnification is applied to obtain the number of solid grains and study the uniform distribution of solid grains. But it is worthy to have the microstructure photo with higher magnification for the investigation of the mechanism of semi-solid microstructural evolution, which will be carried out in our future research.

  1. How was the tensile test performed? This must be specified in the article.

Response: Thanks for the reviewer’s comment. We have added the specific process parameters for the tensile and hardness tests. The added content is shown as follows.

For obtaining the tensile strength and elongation, samples were stretched to break at the speed of 2 mm/min. In order to test the hardness, samples were processed under the test load of 200 g with the pressure holding time of 15 s.

Please see lines 165-167 on page 5 in the revised manuscript.

I believe that the article can be accepted after minor revision.

Response: Thank you for the acknowledgment of our work.

Reviewer 2 Report

Dear Authors, the presented topic is interested but I have some comments:

  1. The title contains "optimisation" of the process, but actually you haven't used any optimisation method, just the selection of process parameters for the best results. The title must be changed or the optimisation method applied with the statistical analysis. 
  2. What is the novelty of the work? EMS is a known method to improve the casting process. 
  3. What is the error type in Figure 6, 9?
  4. There are no errors in Figures 7, 10.
  5. The units in Figures are given in a strange way, should be after comma sign, e.g., "Tensile strength, MPa" or in brackets [MPa]. 
  6. Objects in Figure 12 should be bigger, it is difficult to see anything. 
  7. Celsius degrees are written without the space between the number and units, e.g. 12°C
  8. You concluded, that the best range of the parameters is 595-600°C and 100 MPa, but no other squeeze pressure was selected for the temperature 600 and higher. Why it was limited to 100°C?  How do you know that e.g. for 600°C 100 MPa is the best value, when you haven't checked other values?

Author Response

The authors would like to take this opportunity to sincerely thank the reviewer for his/her valuable comments. We have studied all the comments carefully and have made corresponding corrections that we hope will meet with your approval. The detailed responses to the reviewer’s comments are provided below.

Dear Authors, the presented topic is interested but I have some comments:

  1. The title contains "optimisation" of the process, but actually you haven't used any optimisation method, just the selection of process parameters for the best results. The title must be changed or the optimisation method applied with the statistical analysis. 

Response: Thanks for the reviewer’s comment. This study investigated the effects of two operating parameters, insulation temperature and squeeze pressure, on the microstructure and mechanical properties through experiments. The optimal insulation temperature was first identified by Experiments 1-4. Based on the results of Experiments 1-4, Experiments 5-8 were designed at the insulation temperature of 595℃ to identify the optimal squeeze pressure. The added content is shown as follows.

It is worthy to note that the insulation temperature in Experiments 5-8 are designed based on the results from Experiments 1-4 to only investigate the effects of different squeeze pressure without the variation in the insulation temperature.

According to the analysis in Section 3.2, it was found that the microstructure and mechanical properties of the scroll were close between 595 and 600°C. Therefore, the insulation temperature was designed as 595°C to investigate the effects of the squeeze pressure without the variation in the insulation temperature for the purpose of energy saving.

The relevant information of the optimization method has been added in the revised manuscript. Please see lines 139-142 on page 4 and lines 234-237 on page 8.

  1. What is the novelty of the work? EMS is a known method to improve the casting process. 

Response: Thanks for the reviewer’s comment. The introduction has been revised in order to show the novelty of this work. We agree with the reviewer that EMS is a known method to improve the casting process. In this paper, the EMS based semi-solid multi-cavity squeeze casting (SMSC) process is proposed for manufacturing aluminium alloy scrolls. The proposed process aimed to effectively improve production efficiency, microstructure, and mechanical properties of the scroll simultaneously. The added and revised content is shown as follows.

This work aims to effectively improve production efficiency, microstructure, and mechanical properties of the scroll. Firstly, the near-final forming of the scroll can be realized via the processed process, significantly improving the material utilization rate. Secondly, the concept of one mould for four castings can be verified, obviously improving the production efficiency. Thirdly, the semi-solid billet with fine and spherical grains can be prepared by the EMS process, improving the mechanical properties of the scroll.

Please see the revision lines 87-93 on page 2 in the revised paper.   

  1. What is the error type in Figure 6, 9?

Response: The error type in Figures 6 and 9 are standard deviation. The explanation of it has been added in the revised manuscript. Please see lines 167-168 on page 5.

  1. There are no errors in Figures 7, 10.

Response: Thanks for pointing out this. The error bars have been added in Figures 7 and 10. Please see the revision on pages 8 and 11.

  1. The units in Figures are given in a strange way, should be after comma sign, e.g., "Tensile strength, MPa" or in brackets [MPa]. 

Response: Thanks for pointing this out. All units in the figures have been revised with the comma sign. Please see the revision in Figures 6, 7, 9, 10 in the revised paper.

  1. Objects in Figure 12 should be bigger, it is difficult to see anything. 

Response: Thanks for the reviewer’s comment. Figure 12 has been revised as suggested. Please see line 340 on page 13.

  1. Celsius degrees are written without the space between the number and units, e.g. 12°C

Response: Thanks for pointing out this. The space between the number and units for Celsius degrees have all been deleted in the revised manuscript.

  1. You concluded, that the best range of the parameters is 595-600°C and 100 MPa, but no other squeeze pressure was selected for the temperature 600 and higher. Why it was limited to 100°C?  How do you know that e.g. for 600°C 100 MPa is the best value, when you haven't checked other values?

Response: Thanks for the reviewer’s comment. In this paper, the optimal temperature was first identified under the same pressure of 100 MPa in Experiments 1-4. The insulation temperature in Experiments 5-8 are designed based on the results from Experiments 1-4 to only investigate the effects of different squeeze pressure without the variation in the insulation temperature. As the microstructure and mechanical properties were observed significantly close between 595 and 600°C, the insulation temperature in Experiments 5-8 were designed as 595°C for the purpose of energy saving.

The relevant explanation has been added in the revised manuscript as shown as follows.

It is worthy to note that the insulation temperature in Experiments 5-8 are designed based on the results from Experiments 1-4 to only investigate the effects of different squeeze pressure without the variation in the insulation temperature.

According to the analysis in Section 3.2, it was found that the microstructure and mechanical properties of the scroll were close between 595 and 600°C. Therefore, the insulation temperature was designed as 595°C to investigate the effects of the squeeze pressure without the variation in the insulation temperature for the purpose of energy saving.

Please see lines 139-142 on page 4 and lines 234-237 on page 8.

Reviewer 3 Report

The paper “Experimental investigation and optimisation of the semi-solid multi-cavity squeeze casting process for the wrought aluminum alloy scroll” describes prospective technology for obtaining the thin-walled castings using wrought aluminum alloy. Used by the authors technology of squeeze casting may be applied not only to the “classic” aluminum alloys but also for obtaining casting with good structure and properties from metal-matrix aluminum composites (e.g. https://doi.org/10.1016/j.jmapro.2020.09.067,  http://dx.doi.org/10.1007/s11665-017-2734-3, https://doi.org/10.1016/j.msea.2017.12.091, etc.).  The paper is well written and maybe interesting for the readers. However, some points in the manuscript are questionable and should be modified or described in more details:

  1. The applied by the authors technology may be used not only for the alloys. The authors should analyze in the introduction part the application of the squeeze casting technology for the manufacturing of the parts with complex shapes from aluminum metal-matrix composites. This technology significantly increases the wettability of the particles and decreases porosity.
  2. The liquidus temperature of the investigated alloy is 615ËšC. It seems to be strange why the authors have chosen the insulation temperature for the 1 and 2 experiments as 615 and 645 ËšC, correspondently. At these temperatures, the alloy is out of the semi-solid range and cannot be assessed for the current investigation. The additional explanations should be added to the Materials and Methods part.
  3. How the authors have recognized the α1-(Al) and α2-(Al). On the microstructure we see the cross-sections of the different grains. Most of the grains are not crossed at the center and it is hard to separate different types of grains. It is better to remove from the paper two types of the (Al) grains and analyze only the influence of the different technology parameters on the average grain size.
  4. The main tensile strength property for the constructors is yield stress (YS), but not ultimate strength. The authors should add to the paper analysis of the influence of different technological parameters on the YS. Also, it is recommended for the authors to analyses the influence of the grain size on the YS (for example by using the Hall-Petch relation). It may answer the question is the grain size is the main strengthening effect or maybe others structural parameters also change significantly (dislocation density increases, size of the crystallization origin particles decreases, etc.)
  5. Minor corrections are also required:

- the error (±) of the properties should be added to the values throughout the article. The number of the digits in the values of the structure parameters and properties should be decreased accordingly error value (for example, in the Abstract instead of 110.92 μm should be 111±3);

- error bars should be added in figures 7 and 10.

- information about mechanical test conditions and equipment should be added.

Author Response

The authors would like to take this opportunity to sincerely thank the reviewer for his/her valuable comments. We have studied all the comments carefully and have made corresponding corrections that we hope will meet with your approval. The detailed responses to the reviewer’s comments are provided below.

The paper “Experimental investigation and optimisation of the semi-solid multi-cavity squeeze casting process for the wrought aluminum alloy scroll” describes prospective technology for obtaining the thin-walled castings using wrought aluminum alloy. Used by the authors technology of squeeze casting may be applied not only to the “classic” aluminum alloys but also for obtaining casting with good structure and properties from metal-matrix aluminum composites (e.g. https://doi.org/10.1016/j.jmapro.2020.09.067,  http://dx.doi.org/10.1007/s11665-017-2734-3, https://doi.org/10.1016/j.msea.2017.12.091, etc.).  The paper is well written and maybe interesting for the readers. However, some points in the manuscript are questionable and should be modified or described in more details:

  1. The applied by the authors technology may be used not only for the alloys. The authors should analyze in the introduction part the application of the squeeze casting technology for the manufacturing of the parts with complex shapes from aluminum metal-matrix composites. This technology significantly increases the wettability of the particles and decreases porosity.

Response: Thanks for pointing out this. The introduction has been revised by adding the advantages of squeeze casting for manufacturing complex components from aluminium metal-matrix composites. Relevant references have also been added. The revised and added content is shown as follows.

This gas and shrinkage porosity can be effectively eliminated by the squeeze casting technology, which has been recommended to manufacture the complex component from aluminum alloys and aluminum metal-matrix composites [12-14].

Please see the revision in liens 47-49 on page 2.

  1. The liquidus temperature of the investigated alloy is 615ËšC. It seems to be strange why the authors have chosen the insulation temperature for the 1 and 2 experiments as 615 and 645 ËšC, correspondently. At these temperatures, the alloy is out of the semi-solid range and cannot be assessed for the current investigation. The additional explanations should be added to the Materials and Methods part.

Response: Thanks for pointing this out. The temperatures of 615 and 645ËšC were designed in Experiments 1 and 2, respectively, in order to compare and analyze the advantages of the semi-solid forming process with the near liquidus and liquid forming process. The revised and added content is shown as follows.

In order to compare and analyze the advantages of the semi-solid forming process compared with near liquidus and liquid forming process, the liquid temperature of 645°C, the near liquidus temperature of 615°C, and the semi-solid temperature of 600 and 595°C were designed in Experiments 1-4, respectively, to study the effect of insulation temperature on the microstructures.

Please see lines 132-136 on page 4 in the revised manuscript.

  1. How the authors have recognized the α1-(Al) and α2-(Al). On the microstructure we see the cross-sections of the different grains. Most of the grains are not crossed at the center and it is hard to separate different types of grains. It is better to remove from the paper two types of the (Al) grains and analyze only the influence of the different technology parameters on the average grain size.

Response: Thanks for the reviewer’s comment. Two types of (Al) grains have been removed to better focus on the influence of the operating parameters on the average grain size. Please see the revision in lines 183-184 on page 6, lines 221-222 on page 7, lines 239-243 on page 8, lines 342-343 and 353-354 on page 13, and lines 385-386 on page 15.

  1. The main tensile strength property for the constructors is yield stress (YS), but not ultimate strength. The authors should add to the paper analysis of the influence of different technological parameters on the YS. Also, it is recommended for the authors to analyses the influence of the grain size on the YS (for example by using the Hall-Petch relation). It may answer the question is the grain size is the main strengthening effect or maybe others structural parameters also change significantly (dislocation density increases, size of the crystallization origin particles decreases, etc.)

Response: Thanks for the reviewer’s comment. We agree with the reviewer that the main tensile strength property for the constructors is yield stress and it is worthy to investigate the influence of operating parameters on the yield stress. However, the product studied in this paper is the scrolls for the scroll compressor. The fracture of the scroll is the general damage form of the scroll compressor, which is mainly caused by the significant impact force generated during the operation. Therefore, the ultimate strength is investigated in this study as it can reflect the difficulty of the break occurrence, which is an important indicator for the quality of the scroll.

Relevant explanations have been added in the revised manuscript, which is shown as follows.

The fracture of the scroll is the general damage form of the scroll compressor which is mainly caused by the significant impact force generated during the operation. The ultimate strength is investigated in this study as it reflects the difficulty of the break occurrence, which is an important indicator of the scroll quality. Therefore, the mechanical properties investigated in this study contains the tensile strength, elongation, and hardness.

Please see lines 206-210 on page 7.

  1. Minor corrections are also required:

- the error (±) of the properties should be added to the values throughout the article. The number of the digits in the values of the structure parameters and properties should be decreased accordingly error value (for example, in the Abstract instead of 110.92 μm should be 111±3);

Response: Thanks for the reviewer’s comment. The error (±) has been added to the values in the study and the digits of all numbers have been corrected as suggested. Please see the revision in the revised paper.

- error bars should be added in figures 7 and 10.

Response: Thanks for the reviewer’s comment. Error bars have been added in Figures 7 and 10 as suggested. Please see the revision in line 230 on pages 8 and line 298 on page 11.

- information about mechanical test conditions and equipment should be added.

Response: Thanks for the reviewer’s comment. Information about the mechanical test conditions and equipment has been added in the revised manuscript. The added information is shown as follows.

Samples from the central region of the scroll (Position A2) were used for testing the mechanical properties by using the universal materials tester (INSTRON 5982) and the hardness tester (HV-1000). For obtaining the tensile strength and elongation, samples were stretched to break at the speed of 2 mm/min. In order to test the hardness, samples were processed under the test load of 200 g with the pressure holding time of 15 s.

Please see the revision in lines 163-167 on page 5.

Round 2

Reviewer 2 Report

I accept the explanations. 

Author Response

Comments: I accept the explanations.

Response: Thanks for your time and valuable comments.

Reviewer 3 Report

The authors have mainly improved the paper accordingly to previous comments. However, in my opinion, the yield stress values should be added to the manuscript. The authors wrote that the ultimate strength reflects the difficulty of the break occurrence, which is an important indicator of the scroll quality. However, the presence of even small deformation (less than 1 %) makes the scroll out of work. Therefore, the yield strength should be analyzed in the manuscript accordingly comment #4 from the previous review as it is the rheological property to small deformation.

Author Response

The authors would like to take this opportunity to sincerely thank the reviewer for his/her valuable comments. We have further revised the manuscript according to the reviewer's comments. We hope the revised manuscript will meet with your approval. The detailed responses to the reviewer’s comments are provided below.

Comments: The authors have mainly improved the paper accordingly to previous comments. However, in my opinion, the yield stress values should be added to the manuscript. The authors wrote that the ultimate strength reflects the difficulty of the break occurrence, which is an important indicator of the scroll quality. However, the presence of even small deformation (less than 1 %) makes the scroll out of work. Therefore, the yield strength should be analyzed in the manuscript accordingly comment #4 from the previous review as it is the rheological property to small deformation.

Response: Thanks for the reviewer’s comment. The manuscript has been revised by adding the data of the yield strength in Figure 7, Figure 10, and Table 3. The analysis of the yield strength results has been added in Section 3. Relevant content in the manuscript has also been revised. The revised and added content is shown as follows.

Lines 24-25 on page 1: The maximum tensile strength, yield strength, and hardness were observed as 386±8 MPa, 228±5 MPa, and 117±5 HV, respectively, at the squeeze pressure of 100 MPa.

Lines 162-163 on page 5: For obtaining the tensile strength, yield strength, and elongation, samples were stretched to break at the speed of 2 mm/min.

Lines 205-206 on page 7: The mechanical properties investigated in this study contains the tensile strength, yield strength, elongation, and hardness.

Lines 208-209 on page 7: It was found that the tensile strength, yield strength, elongation, and hardness all increased and almost stabilized when the insulation temperature dropped from the liquid state to the semi-solid temperature range.

Lines 214-216 on page 7: An increase in the yield strength was observed from 163±13 MPa at the insulation temperature of 645°C to 228±5 MPa at the insulation temperature of 595°C.

Lines 225-229 on page 7: The improvement in the yield strength with the decrease in the insulation temperature was because of the reduced grain size, which can be expressed by the classical Hall-Petch relationship as shown in Equation (3)[33,34]

Figure 7 on page 8. 

Lines 286-290 on page 10: The yield strength increased from 214±5 MPa at the squeeze pressure of 25 MPa to 228 ±5 MPa at the squeeze pressure of 100 MPa, which dramatically decreased to 212±5 MPa when the semi-solid billet is processed at the squeeze pressure of 125 MPa. This can be attributed to the change in the average grain size as the squeeze pressure rose.

Figure 10 on page 11:

Lines 317-318 on page 12: It was found that both the hardness and, tensile strength, and yield strength of the scroll after the T6 heat treatment were higher than those without the T6 treatment.

Lines 320-321 on page 12: The tensile strength was enhanced to 386±8 MPa while the yield strength was increased to 228±5 MPa through the T6 heat treatment.

Table 3 on page 12.

Lines 377-378 on page 14: The maximum tensile strength, yield strength, and hardness were observed as 386±8 MPa, 228±5 MPa, and 117±5 HV, respectively, at the squeeze pressure of 100 MPa.
